# Generation and Characterization of Cisplatin-Resistant Oral Squamous Cell Carcinoma Cells Displaying an Epithelial–Mesenchymal Transition Signature

**DOI:** 10.3390/cells14171311

**Published:** 2025-08-24

**Authors:** Everton Freitas de Morais, Lilianny Querino Rocha de Oliveira, Cintia Eliza Marques, Fábio Haach Téo, Gisele Vieira Rocha, Camila Oliveira Rodini, Clarissa A. Gurgel, Tuula Salo, Edgard Graner, Ricardo D. Coletta

**Affiliations:** 1Graduate Program in Oral Biology, School of Dentistry, University of Campinas, Piracicaba 13414-018, SP, Brazil; l265902@dac.unicamp.br (L.Q.R.d.O.); 207099@dac.unicamp.br (C.E.M.); coletta@fop.unicamp.br (R.D.C.); 2Department of Oral Diagnosis, School of Dentistry, University of Campinas, Piracicaba 13414-018, SP, Brazil; fabioteo@unicamp.br (F.H.T.); graner@unicamp.br (E.G.); 3Gonçalo Moniz Institute, Oswaldo Cruz Foundation (IGM-FIOCRUZ/BA), Salvador 40296-710, BA, Brazil; givieiraroch@gmail.com (G.V.R.); clarissa.gurgel@fiocruz.br (C.A.G.); 4Department of Biological Sciences, Bauru School of Dentistry, University of São Paulo, Bauru 05508-220, SP, Brazil; carodini@usp.br; 5Department of Propaedeutics, School of Dentistry, Federal University of Bahia, Salvador 40110-909, BA, Brazil; 6Cancer and Translational Medicine Research Unit, Faculty of Medicine and Medical Research Center Oulu, Oulu University Hospital, University of Oulu, 90014 Oulu, Finland; tuula.salo@oulu.fi; 7Department of Pathology, Helsinki University Hospital, Institute of Oral and Maxillofacial Disease, University of Helsinki, and HUSLAB, 00029 Helsinki, Finland

**Keywords:** oral squamous cell carcinoma, cisplatin resistance, tumor plasticity, epithelial–mesenchymal transition, biomarkers

## Abstract

Cisplatin resistance remains a major therapeutic challenge in oral squamous cell carcinoma (OSCC), leading to treatment failure and poor outcomes. This study aimed to generate and characterize cisplatin-resistant OSCC models to elucidate resistance mechanisms. Two resistant OSCC cell lines (SCC-9R and HSC-3R) were developed through gradual dose escalation. Parental and resistant cells were analyzed via RNA-seq and gene set enrichment analysis, and validated through RT-qPCR, Western blot, immunofluorescence, and gelatin zymography. Functional assays, including 2D and 3D migration and invasion models, assessed phenotypic changes. A multi-omics analysis revealed molecular alterations in resistant cells, including 305 differentially expressed genes (DEGs) in HSC-3R (187 upregulated) and 782 in SCC-9R (298 upregulated) versus parental lines, with enrichment for extracellular matrix organization (*p* < 0.001) and consistent epithelial–mesenchymal transition (EMT) activation (*p* < 0.001), demonstrated by the upregulation of ZEB1, ZEB2, Vimentin, and TWIST1, and E-cadherin suppression. Functional validation confirmed an aggressive phenotype, including increased migration (*p* < 0.05), invasion (*p* < 0.01), and elevated MMP-2 (*p* < 0.01) and MMP-9 (*p* < 0.001) activity. Findings were verified in 3D spheroid models. Overall, cisplatin resistance in OSCC involves EMT, inflammatory signaling, and metabolic adaptation. The consistency of these features across both models supports the robustness of this in vitro system and reveals targets for therapeutic intervention.

## 1. Introduction

Over 380,000 cases of oral cavity cancer are diagnosed each year, with oral squamous cell carcinoma (OSCC) accounting for at least 90% of these cases [1,2]. Despite significant efforts to reduce OSCC-related morbidity and mortality, the five-year overall survival rate remains unacceptably low [2].

Surgery remains the primary treatment for OSCC, often complemented by adjuvant radiotherapy and chemotherapy in cases at an advanced stage [3]. Radiotherapy and/or chemotherapy may also serve as the primary treatment options, particularly for patients who are not candidates for surgical intervention [4]. Cisplatin, a DNA-damaging agent, is a widely used anticancer drug in chemotherapy for various human cancers, including OSCC [5]. However, resistance to cisplatin, which can be observed in up to 30% of patients [6], grants cancer cells tolerance to its effects [7,8]. The acquisition of resistance not only reduces treatment efficacy but also tends to generate tumor cells towards increased aggressiveness, leading to a poorer prognosis for patients [9,10]. Given that cisplatin resistance poses a major challenge to OSCC treatment and patient survival, it is of the utmost importance to understand the mechanisms that regulate it, as well as the discovery of agents that restore cisplatin sensitivity for improving therapeutic outcomes.

The epithelial–mesenchymal transition (EMT) has emerged as a key driver of therapy resistance [11]. As a highly conserved cellular program, EMT enables immobile and polarized epithelial cells to transform into motile cells with mesenchymal-like features such as the loss of apicobasal polarity, disruption of cell–cell junctions, actin cytoskeleton reorganization, and increased capacity to invade the extracellular matrix as an individual cell [12]. EMT is linked to tumor progression, metastasis, and resistance to conventional therapies [12,13,14]. However, the precise mechanisms by which EMT induces drug resistance remain under investigation and may vary among different types of cancer. For instance, studies suggest that EMT shifts tumors from a proliferative state to a quiescent state, promoting resistance to immunotherapy [15,16,17]. Additionally, reversing EMT has been shown to restore sensitivity to anticancer drugs [18]. EMT often generates cells with stem-like properties [19,20], which exhibit resistance to apoptosis and other forms of programmed cell death with the maintenance of an enhanced self-renewal capacity [21].

In this study, we generated and characterized two cisplatin-resistant OSCC cell lines (SCC-9R and HSC-3R), establishing a valuable model for investigating the molecular pathways and potential markers of EMT linked to resistance in oral cancer.

## 2. Materials and Methods

### 2.1. Cell Culture

SCC-9 (ATCC, Manassas, VA, USA) and HSC-3 (JCRB 0623; Osaka National Institute of Health Sciences, Osaka, Japan) cells were cultured in DMEM and Ham’s F12 medium (DMEM/F12; Gibco™, Carlsbad, CA, USA) containing 10% fetal bovine serum, 400 ng/mL hydrocortisone (Sigma-Aldrich, St. Louis, MO, USA), and antibiotics (Pen Strep, Gibco™, Carlsbad, CA, USA) at 37 °C in an incubator at 37 °C with 5% CO_2_. A cancer-associated fibroblasts (CAFs) cell line was established, cultured, and characterized as previously described [22]. Cells were regularly tested for mycoplasma contamination (MycoAlert® Mycoplasma Detection Kit, Lonza, Basel, Switzerland).

### 2.2. 3D Spheroid Model

Cells were cultured in 3D spheroids as previously described [23]. For viability assays, spheroids were established with 10,000 cells, and spheroids with 30,000 cells were constructed for migration and invasion assays.

### 2.3. Drug

Cisplatin [cis-diammineplatinum(II) dichloride] was obtained from Sigma-Aldrich (St. Louis, MO, USA) and dissolved in 0.9% NaCl. Aliquots were stored at −20 °C and thawed immediately before use.

### 2.4. Induction of Cisplatin-Resistance in OSCC Cells

Cisplatin-resistant (HSC-3R and SCC-9R) cells were derived from their respective parental lines (HSC-3P and SCC-9P) through continuous cisplatin exposure, following the determination of the IC_50_ values from the initial dose–response curves. Each line was treated with cisplatin in progressive doses (IC_2.5_, IC_5_, IC_10_, IC_20_, IC_25_, and IC_50_) for 24 h. Cells were exposed for at least three cycles of treatment at each dosage before progression to a higher dosage. Following each treatment cycle, the cisplatin medium was removed, and the cells were allowed to recover for a minimum of 48 h before being re-exposed to the treatment. After the development period, IC_50_ concentrations were reassessed in each resistant line. Cells were continuously maintained in cisplatin at the IC_5_ concentration, with the drug removed before the assays. This process can be visualized in Appendix A.

### 2.5. Drug Sensitivity Assay

Cells (8000 cells per well) were plated in quadruplicate in wells of a 96-well culture plate (Corning Inc., Corning, NY, USA) and cultured at 37 °C for 24 h. To evaluate the sensitivity, cisplatin at different concentrations (0.3125 to 400 μM) was added to the culture medium and incubated for 24 h. Viability was also verified with cells cultured in spheroids (10,000 cells per spheroid). The spheroids were treated with cisplatin ranging from 0.3125 to 480 μM for 24 h, with the IC_50_ subsequently calculated. Cell viability was determined by CellTiter-Glo 3D Cell Viability (Promega, Madison, WI, USA) according to the manufacturer’s protocol.

### 2.6. Morphology and Organization of Cytoskeleton

To detect potential changes in the morphology and organization of the cell cytoskeleton following the induction of chemoresistance, fluorescent staining with phalloidin (Alexa Fluor™ 488 Phalloidin, Invitrogen, Carlsbad, CA, USA) and nuclear staining with DAPI (Invitrogen, Carlsbad, CA, USA) were performed. Images were acquired using confocal microscopy (Leica Microsystems, Wetzlar, Germany).

### 2.7. RNA Extraction, Library Preparation, and Sequencing

RNA sequencing (RNA-seq) was performed on parental and chemo-resistant cells in independent triplicates. After total RNA extraction, quality analysis, and quantification, RNA libraries were prepared using a poly(A) enrichment kit (Illumina, San Diego, CA, USA). Libraries were quantified, and pools prepared for multiplexed sequencing. Subsequently, clustering and sequencing were performed on the Illumina platform with 100 bp paired-end (PE) reads and an average coverage of 30 million reads per sample.

The FASTQ files were pre-processed to generate count matrices. The pipeline comprised the following: quality control, trimming of low-quality reads, alignment to the reference genome, and gene expression quantification. Initial sequence quality was assessed using FastQC (v0.11.9; Babraham Bioinformatics, Cambridge, UK), which provided metrics such as per-base quality, GC content, and adapter contamination. Low-quality reads were trimmed using Trimmomatic (v0.39; Usadel Lab, Aachen, Germany) with the TRAILING:10 parameter, removing bases with a Phred score below 10 at the 3′ end. Post-trimming, FastQC was re-run to verify quality improvements. Filtered reads were aligned to the GRCh38 human reference genome using HISAT2 (v2.1.0; Johns Hopkins University, Baltimore, MD, USA) with the –rna-strandness R option for reverse-stranded single-end reads. The resulting alignments were sorted and converted to the BAM format using samtools sort. Finally, gene-level read counts were obtained with featureCounts (v2.0.0; Walter and Eliza Hall Institute of Medical Research, Melbourne, Australia), using the gene annotation file Homo_sapiens.GRCh38.106.gtf.

### 2.8. Data Processing and Identification of Differentially Expressed Genes (DEGs)

Gene expression counts were obtained from files generated by featureCounts. Raw data files were loaded and processed using R (v4.3.0; R Foundation for Statistical Computing, Vienna, Austria). The read_in_feature_counts function was defined to read the count files. For gene annotation, a GTF file (GRCh38.p14) from GENCODE (www.gencodegenes.org/human/release_46.html, accessed on 15 November 2024) containing gene information was used. The file was loaded using the rtracklayer package and filtered to select only entries of type “gene.” The annotated information was used to associate gene identifiers with the count data. The count data were then organized into a DESeqDataSet object using the DESeq2 package (Bioconductor, Roswell Park Comprehensive Cancer Center, Buffalo, NY, USA), with the experimental design structured according to OSCC cell lines, including resistant (SCC-9R and HSC-3R) and parental (SCC-9P and HSC-3P) groups. From the DESeqDataSet object, genes were annotated by matching their identifiers with the GTF file. Unannotated genes were removed. Only genes classified as protein-coding were retained for further analysis.

The data were normalized by estimating size factors using the estimateSizeFactors function. To ensure quality, low-expression genes were filtered out, retaining only those with a normalized count greater than 10 in at least three samples. The normalized data were transformed using variance-stabilizing transformation (VST) to reduce the influence of technical variability and improve suitability for an exploratory analysis and visualization. A differential expression analysis was performed by estimating gene-wise dispersions with the estimateDispersions function. DEGs between resistant and parental cell lines were identified using the results function in DESeq2 [24], applying independent filtering with a significance threshold of alpha = 0.05, a log2 fold change threshold of 1.5, and Benjamini–Hochberg-adjusted *p*-values (pAdjustMethod = “BH”).

### 2.9. Quantitative RT-PCR (RT-qPCR) and Western Blot

Total RNA was extracted from cell lines using TRIzol™ reagent (Invitrogen, Carlsbad, CA, USA) and converted into cDNA using the SuperScript™ IV First-Strand Synthesis System (Invitrogen, Carlsbad, CA, USA). Expression levels were quantified on the StepOnePlus™ Real-Time PCR System (Applied Biosystems, Foster City, CA, USA) using SYBR® Green PCR Master Mix (Applied Biosystems, Foster City, CA, USA). Cyclophilin A (PPIA) was used as the housekeeping reference gene in the amplification, with relative expression quantified using the 2^−ΔΔCt^ method. Detailed information on all primers is available in Appendix A. A Western blot analysis was performed according to de Morais et al. [14]. Full details of Western blot, including the list of antibodies, are provided in Appendix A.

### 2.10. Immunofluorescence

Cells were cultured on culture chamber slides (Lab-Tek™, Thermo Fisher Scientific, Rochester, NY, USA) and fixed with 70% (*v*/*v*) ethanol (Sigma-Aldrich, St. Louis, MO, USA) in phosphate-buffered saline (PBS) for 60 min at room temperature. Cells were then incubated with primary antibodies diluted in PBS containing 1% bovine serum albumin (BSA) (Sigma-Aldrich, St. Louis, MO, USA) for 2 h at room temperature, followed by fluorophore-conjugated secondary antibodies in the dark for 1 h. Confocal microscopy (Leica Microsystems, Wetzlar, Germany) set at 370, 488, and 543 nm for DAPI (Invitrogen, Carlsbad, CA, USA), Alexa Fluor^®^ 488, and Alexa Fluor^®^ 555 fluorescence, respectively, was utilized. Images were exported as 16-bit TIFF files. In Appendix A, all antibodies used in this analysis are presented.

### 2.11. Gelatin Zymography

A zymographic analysis was performed as previously described [25]. Briefly, conditioned media of the cells were submitted to electrophoresis on 10% SDS-PAGE gels containing 1 mg/mL gelatin under nonreducing conditions to evaluate MMP-2 and MMP-9. Gels were stained with Coomassie blue, and gelatin digestion was identified as clear bands against the blue background. Following image capture, the intensities of the negative bands were analyzed using the Molecular Analysis software, version 9.7 (UVITEC Ltd., Cambridge, UK).

### 2.12. Migration and Invasion Assays

The 2D cell migration and invasion assays, along with the 3D invasion assay, were conducted following the methodology described by Dourado et al. [23]. For the 2D co-culture migration and invasion assays, a 4:1 ratio of cancer cells to CAFs was employed. The migration assay was also assessed by a scaffold-free 3D spheroid model. For this assessment, the spheroids were individually transferred to wells of an adherent 96-well plate to ensure adhesion to the bottom of the plate. The spheroids were treated in serum-free culture medium, and cell migration from the spheroids was monitored. Images of the spheroids were captured immediately and again after 48 h of culture. Cancer cell migration in each spheroid was quantified by measuring the mean distance across 16 migration hotspots, defined as the farthest distance of cancer cells from the center of the tumor spheroid.

### 2.13. Statistical Analysis

The data represent the means ± SD, and all assays were performed at least three times. Data were assessed for normality (Gaussian distribution), and appropriate statistical tests were applied, including the Mann–Whitney test or Student’s *t*-test. The level of significance was settled at 5% (*p* ≤ 0.05). Analyses were performed using GraphPad Prism version 8 (Dotmatics, Boston, MA, USA).

## 3. Results

### 3.1. Establishment of Cisplatin-Resistant Cells

The chemoresistance induction lasted approximately 6 months for SCC-9R and 10 months for HSC-3R. Following the process, dose–response curves were generated to determine the new IC_50_ values under 2D and 3D culture conditions. A significant increase in the IC_50_ was observed in these cells compared to the parentals (SCC-9P and HSC-3P) (Figure 1A–D). The phase–contrast morphology analysis after DAPI/Phalloidin staining revealed that resistant cells acquired an elongated, spindle-shaped morphology with dendritic processes, consistent with mesenchymal transition (Figure 1E–H).

### 3.2. Pathway Enrichment and EMT-Related Gene Expression in Cisplatin-Resistant Cells

The gene expression matrix generated from the RNA-seq data processing of parental (HSC-3P and SCC-9P) and resistant (HSC-3R and SCC-9R) cells comprised 13,350 genes. A differential expression analysis between HSC-3P and HSC-3R identified 305 DEGs, with 187 upregulated and 118 downregulated in the resistant cells (Figure 2A). The comparison between SCC-9P and SCC-9R demonstrated 782 DEGs, including 298 upregulated and 484 downregulated in resistant cells (Figure 2B).

A functional enrichment analysis of DEGs from parental-resistant comparisons was performed using the g:Profiler server (https://biit.cs.ut.ee/gprofiler/gost, accessed on 15 November 2024). Detailed results, including complete lists of enriched GO terms and KEGG pathways, along with associated genes, adjusted *p*-values, and fold enrichment values, are provided in Appendix A. Resistant cells showed enrichment for biological processes including extracellular matrix reorganization, collagen metabolism, cell adhesion, positive regulation of MAPK cascade, and lipid transport regulation (Appendix A). HSC-3P exhibited further enrichment in anatomical structure morphogenesis and actin filament organization (Appendix A), and SCC-9P showed further enrichment in cell junction organization, homeostatic processes, and the negative regulation of peptidase activity (Appendix A).

A Gene Set Enrichment Analysis (GSEA) using the GSVA package (v2.7.0) with hallmark gene sets revealed distinct pathway activation patterns between resistant and parental cells. HSC-3R exhibited increased activity in innate immune/inflammatory pathways (complement system, coagulation, TNF-α signaling via NF-κB) and glycolysis, suggesting an inflammatory, metabolically reprogrammed phenotype. In contrast, HSC-3P showed stronger activation of Wnt/β-catenin signaling, which represents pathways linked to epithelial maintenance (Figure 2C). In SCC-9R cells, the increased activation of EMT, downregulation of the UV response, and activation of Hedgehog signaling pathways were observed, consistent with a plastic, stress-adaptive phenotype. SCC-9P cells showed enrichment in proliferative and metabolic pathways (including PI3K/AKT/mTOR signaling, mTORC1 signaling, and MYC targets), along with stress–response pathways (cholesterol homeostasis and reactive oxygen species pathway) (Figure 2D). Together, these findings support that resistant cells preferentially activate pathways related to key hallmarks of therapeutic resistance, such as cellular plasticity, inflammation, and metabolic reprogramming.

Given the GSEA results implicating EMT in resistance, we specifically analyzed canonical EMT markers (Figure 2E,F). Both HSC-3R and SCC-9R showed significantly elevated expressions of mesenchymal regulators, such as ZEB1, ZEB2, and Vimentin, with TWIST1 exclusively upregulated in SCC-9R. TWIST1 expression was also increased in HSC-3R, though it did not reach the statistical threshold for differential expression statistical significance as a DEG (Appendix A). Although not meeting the DEG threshold, SNAIL2 expression was higher in both resistant cells, while CLDN4 was more expressed in parental cells (HSC-3P and SCC-9P), suggesting retained epithelial features (Appendix A).

### 3.3. Validation of the EMT Profile of Cisplatin-Resistant Cells

To evaluate the role of EMT in the chemoresistance induction, we assessed the expression of EMT markers by RT-qPCR, Western blot, and immunofluorescence. Chemoresistance was accompanied by a decrease in E-cadherin expression and a marked increase in the expression of mesenchymal markers Vimentin and N-cadherin, as well as the EMT-inducing transcription factors SNAIL1 and TWIST1. In both resistant cells, we observed a decreased expression of SNAIL2 (SLUG), mainly in SCC-9R cells (Figure 3A,B). These findings were confirmed at the protein level by a Western blot analysis, showing a significant reduction in E-cadherin and increased levels of the mesenchymal marker Vimentin, in synergy with the nuclear transcription factors TWIST1 and SNAIL1 (Figure 3C).

An immunofluorescence assessment endorsed the evident acquisition of the EMT profile, with a significant increase in the expression of N-cadherin, Vimentin, and TWIST1 in the resistant cells (particularly in SCC-9R), as well as the loss of E-cadherin expression (Figure 4).

To evaluate the MMP gene and protein expressions, RT-qPCR and gelatin zymography were employed. Both resistant cells expressed significantly more MMP-2 and MMP-9 transcripts than in the parental cells (Figure 3). To confirm these findings, gelatin zymography was performed showing gelatinolytic activities at molecular weights of ~92 kDa and ~72 kDa in the supernatants of the cells, corresponding to the MMP-9 and MMP-2, respectively. Quantification of these active bands confirmed a significant increase in the amount of MMP-2 in both resistant cells compared to their parental counterparts, as well as an increased amount of MMP-9 in SCC-9R cells (Figure 5).

### 3.4. Chemoresistant Cells Exhibit Enhanced Migratory and Invasive Capacities

The transwell assay revealed significant increased migration and invasion abilities in resistant cells compared to their parental counterparts (Figure 6). The migration of HSC-3R cells increased by 42% (*p* < 0.05), whereas that of SCC-9R was approximately 45% (*p* < 0.01), compared to the corresponding parental cells. The invasion capacity of the extracellular matrix was also higher in HSC-3R (~28% increase, *p* < 0.05) and SCC-9R (~40% increase, *p* < 0.01) cells compared to their parental counterparts. Additionally, in the 3D models, the invasion rate of HSC-3R cells rose by 50% (*p* < 0.005), while SCC-9R cells exhibited an increase of approximately 130% (*p* < 0.0001) (Figure 7). In co-culture with CAFs, resistant cells exhibited even greater migratory and invasive potential. As shown in Figure 6A, the migration of HSC-3R cells increased by approximately 90% (*p* < 0.0001) and SCC-9R cells by approximately 390% (*p* < 0.0001), revealing a significant induction compared to the parental cells.

## 4. Discussion

Although cisplatin remains one of the most widely used chemotherapeutic agents for OSCC, its clinical effectiveness is frequently hindered by the development of drug resistance and dose-dependent toxic side effects [26]. Its cytotoxicity primarily arises from the formation of DNA adducts, which induce genomic damage, stall DNA replication, and trigger cell cycle arrest, ultimately triggering apoptosis in neoplastic cells [8,26]. While research on cisplatin-resistant models in oral cancer has been conducted [27,28], further studies are needed to deepen our understanding of their development and characteristics.

In this study, novel cisplatin-resistant cells were established and characterized, laying the foundation for investigations into mechanisms of cisplatin resistance and therapeutic approaches to overcome resistance in OSCCs. The cisplatin-resistant cells exhibited typical morphological features, including spindle-shaped cells, along with greater variability in cell size compared to their parental counterparts. Additionally, the cisplatin-resistant cells demonstrated significantly higher IC_50_ values, confirming their increased resistance to cisplatin. The observed morphological changes are indicative of the EMT process, previously reported in chemo-resistant lines [27]. The RNA-seq analysis revealed that resistant OSCC cells (HSC-3R and SCC-9R) exhibit distinct molecular adaptations that may underlie cisplatin evasion. Notably, the upregulation of extracellular matrix remodeling genes, activation of inflammatory pathways (e.g., TNFA/NF-κB), and metabolic reprogramming (e.g., enhanced glycolysis) suggest a shift toward a pro-survival, mesenchymal-like phenotype. Furthermore, the marked overexpression of EMT regulators (ZEB1, ZEB2, Vimentin) in resistant cells aligns with their increased plasticity and potential for drug tolerance. These findings suggest that cisplatin resistance in OSCC may be driven not only by classical DNA repair mechanisms but also by broader transcriptional rewiring.

Resistance to cisplatin is not caused by a single factor, and various mechanisms may be involved, with EMT being prominent among them [28]. The overarching mechanism regarding drug resistance associated with EMT is linked to increased drug efflux, reduced cellular proliferation capacity, and the evasion of apoptosis signaling pathways [29]. Furthermore, evading the immune response is another critical mechanism by which EMT contributes to therapeutic resistance, inducing immune escape [30,31,32]. The results of this study also demonstrated significantly increased levels of TWIST1, N-cadherin, Vimentin, MMP-2, and MMP-9, in contrast to significantly decreased levels of E-cadherin in resistant cells compared to parental ones. This finding aligns with studies that have observed that EMT-inducing transcription factors regulate EMT and mediate the development of resistance in neoplastic cells to platinum-based anticancer drugs in various cancer types [29,33,34,35]. Interestingly, we did not observe a significant difference in SNAIL1 expression between parental and resistant cells. In a recent study, Youssef et al. [36] observed that SNAIL1 acts as a pioneering transcription factor inducing EMT in tumor cell lines but does not show significant expression changes between hybrid (partial EMT) and more advanced-stage (complete EMT) cell populations. Collectively, these findings indicate that although SNAIL1 is a crucial nuclear transcription factor initiating EMT, it may not play a major role in sustaining the later stages of EMT, as observed in the resistant cells in our study.

Numerous studies have demonstrated that metastases are governed by phenotypic transitions between epithelial and mesenchymal states (EMT and MET), collectively referred to as epithelial plasticity [29,37,38,39,40,41,42]. Tumor cells can also attain a state of partial or intermediate EMT during the transition between EMT and MET, known as the epithelial/mesenchymal hybrid phenotype. Because of these events, the potential for migration and invasion is modulated, increasing during the transition to EMT [43,44]. The results of this study reveal that the SCC-9R and HSC-3R cells exhibit higher invasion and migration capabilities compared to their respective parental cells. One of the inducing mechanisms for increased migration and invasion capacity in cells undergoing EMT is the upregulation of MMPs [45]. MMP-2 and MMP-9 are enzymes that play a crucial role in extracellular matrix regulation and have been implicated in the process of tumor migration and invasion [46]. The increased expression of MMPs 2 and 9 acts by breaking down structural components of the extracellular matrix, such as collagen, fibronectin, elastin, and proteoglycans [47], creating a more permissive environment for cellular tumor migration and invasion [46,47]. Therefore, reversing EMT in chemo-resistant cells could be an effective method to inhibit the migration and invasion process of neoplastic cells.

Future studies aiming to validate the in vitro findings in animal models may provide a more comprehensive understanding of the behavior of chemo-resistant cells within the context of a living organism. Additionally, it will be crucial to delve deeper into the molecular pathways associated with cisplatin resistance in chemo-resistant cells and identify potential therapeutic methods targeting the reversal of EMT-induced chemoresistance in OSCC cell lines. Functional studies using the targeted modulation of EMT-related factors will be important to establish a direct causal relationship between EMT and chemoresistance, providing direct functional validation of the contribution of EMT to drug resistance and supporting the development of more effective treatments.

## 5. Conclusions

In conclusion, the findings demonstrate the successful establishment of cisplatin-resistant human OSCC cells, highlighting the activation of the EMT process as a critical step in the development of the resistance. These cells serve as valuable models for further investigations into strategies for reversing cisplatin resistance in OSCC, and for identifying the molecular mechanisms underlying chemoresistance.

## Figures and Tables

**Figure 1 cells-14-01311-f001:**
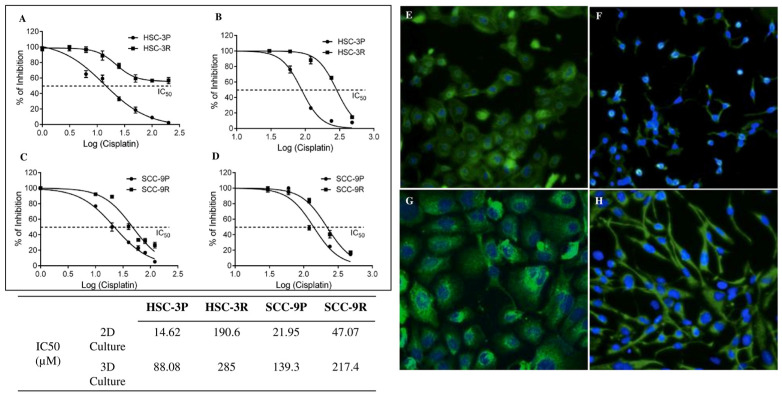
Determination of the IC_50_ concentrations for cisplatin in parental and resistant cells. Curves for HSC-3P and HSC-3R cells in 2D culture and in 3D spheroid culture are depicted in (**A**) and (**B**), respectively. (**C**) The IC_50_ curve for SCC-9P and SCC-9R cultured in monolayer (2D culture), and (**D**) the IC_50_ curve for SCC-9P and SCC-9R cultured in 3D spheroid. Fluorescence microscopy images of Phalloidin/DAPI staining of the parental and resistance cells at original magnification ×200. (**E**) HSC-3P, (**F**) HSC-3R, (**G**) SCC-9P, and (**H**) SCC-9R. All experiments were performed in triplicate and repeated at least three times. Cell viability in both 2D and 3D models was assessed using the CellTiter-Glo 3D Cell Viability assay (Promega, Madison, WI, USA) according to the manufacturer’s protocol.

**Figure 2 cells-14-01311-f002:**
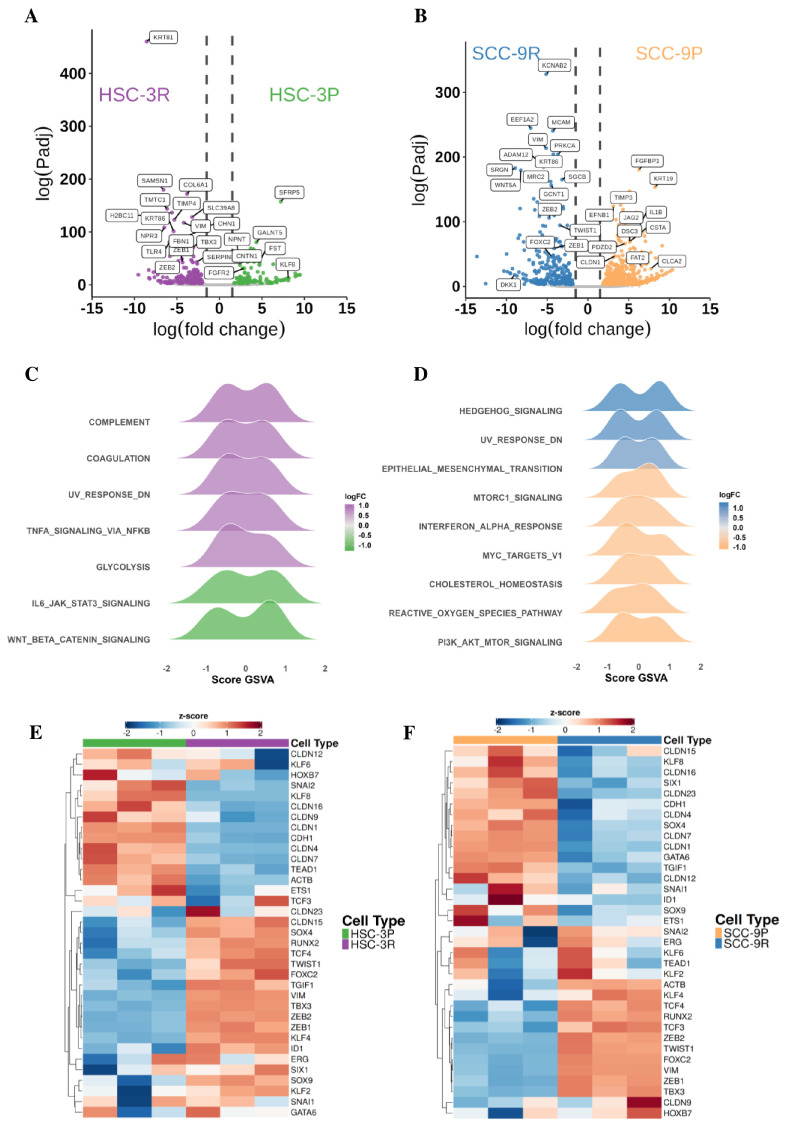
Identification of key genes in OSCC-resistant cells. RNA-seq volcano plot for differentially expressed genes in HSC-3P and HSC-3R cells (**A**) and in SCC-9P and SCC-9R cells (**B**). Gene set variation analysis (GSVA) for the main pathways activated in HSC-3R compared to HSC-3P (**C**) and in SCC-9R compared to SCC-9P (**D**). Heatmap of differentially expressed genes related to epithelial–mesenchymal transition in HSC-3 cells (**E**) and in SCC-9 cells (**F**).

**Figure 3 cells-14-01311-f003:**
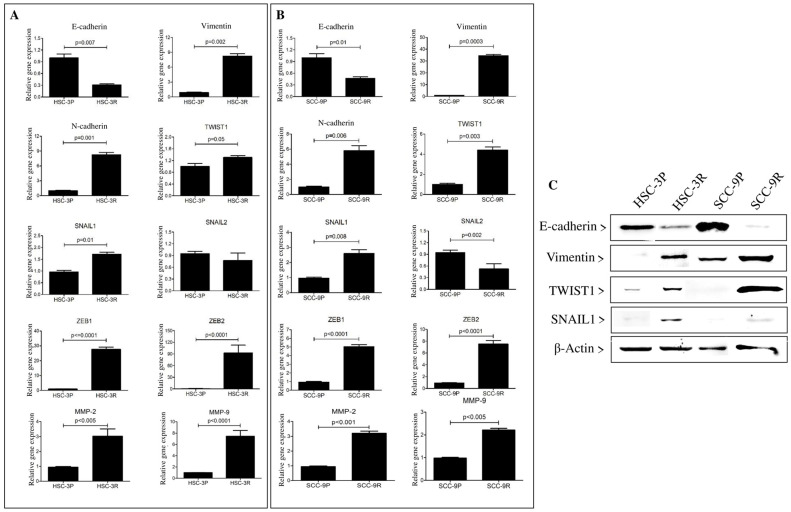
The acquisition of cisplatin resistance triggers the activation of the EMT process in OSCC cells. (**A**,**B**) Quantitative expression analysis of EMT-related markers, including E-cadherin, Vimentin, TWIST1, N-cadherin, SNAIL1, SNAIL2, ZEB1, ZEB2, MMP-2, and MMP-9 mRNA analysis in HSC-3P/R (**A**) and SCC-9P/R (**B**). The values represent the mean ± SD of three separate experiments. The epithelial–mesenchymal shift is more pronounced in SCC-9R cells. Original magnification: ×200. (**C**) Western blot assay for E-cadherin, Vimentin, TWIST1, and SNAIL1 in parental and cisplatin-resistant cells. The reaction against β-actin was used as a housekeeping control. A representative blot is shown. Uniform linear adjustments to brightness and contrast were applied to the entire images for clarity, without altering the relative intensity, shape, or proportion of the bands.

**Figure 4 cells-14-01311-f004:**
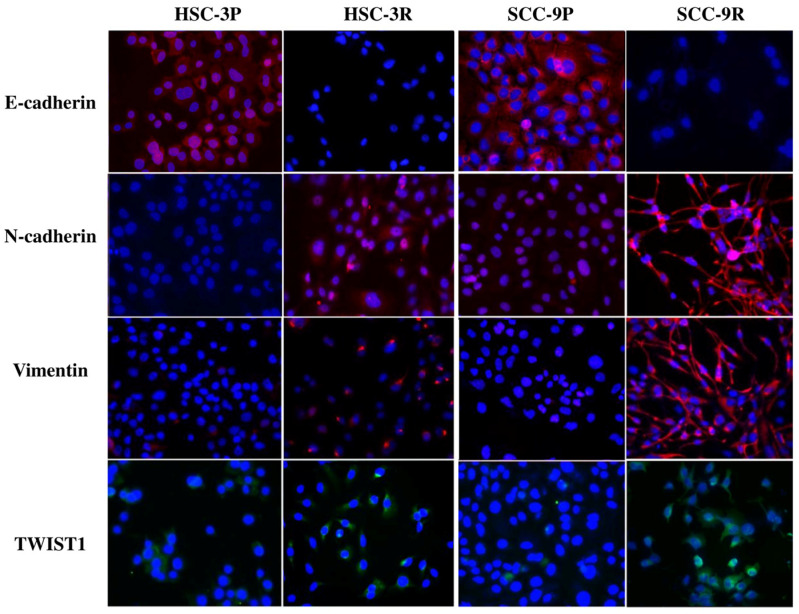
Immunofluorescence analysis was conducted to examine epithelial–mesenchymal transition (EMT) markers in both parental and resistant cells. The images above illustrate an increase in the expression of mesenchymal markers, including N-cadherin, Vimentin, and TWIST1, along with a corresponding loss of E-cadherin expression (original magnification ×100).

**Figure 5 cells-14-01311-f005:**
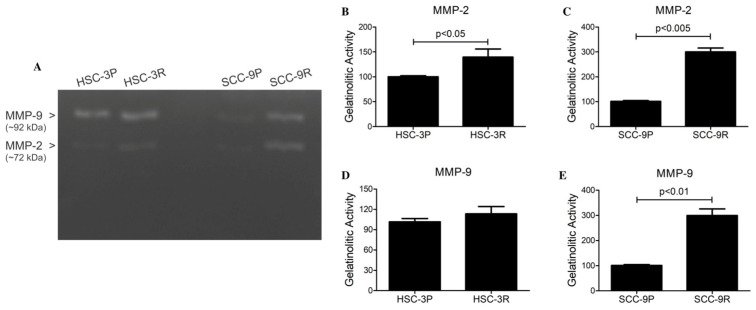
Zymographic analysis of supernatants of parental and resistant cells. (**A**) Representative image of the assay. Zones of gelatinolytic activities were detected at ~92 kDa and ~72 kDa, consistent with the presence of members of the gelatinase family MMP-9 and MMP-2, respectively. Optical densitometric quantification revealed significantly higher amounts of MMP-9 and MMP-2 in SCC-9R compared to SCC-9P, whereas there was only more of MMP-2 in HSC-3R cells. Values are expressed as the mean of optical density units relative to parental cells ± SD of 3 independent experiments. (**B**) MMP-2 quantification in HSC-3 cells, (**C**) MMP-2 quantification in SCC-9 cells, (**D**) MMP-9 quantification in HSC-3 cells, and (**E**) MMP-9 quantification in SCC-9 cells.

**Figure 6 cells-14-01311-f006:**
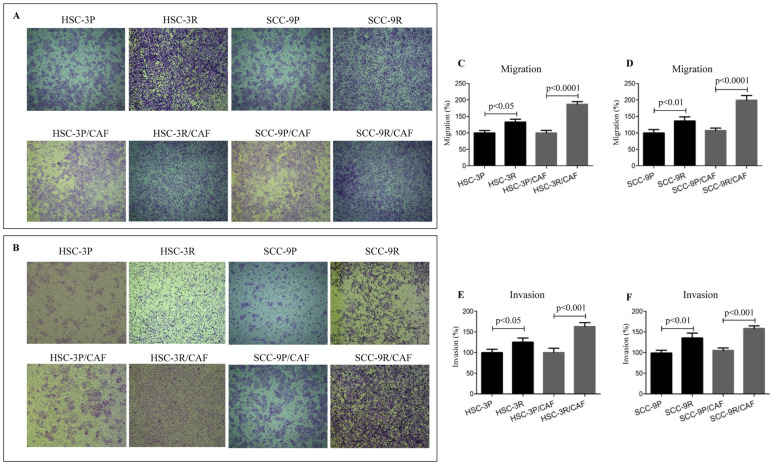
Cisplatin resistance is associated with increased potential for migration and invasion of OSCC cells. (**A**) Representative image for the transwell cell migration assay, and (**B**) the representative transwell cell invasion assay, using Myogel to cover the membrane pores of the inserts. Assays were performed with tumor cells alone or in combination with cancer-associated fibroblasts (CAFs). Quantification of the migrated HSC-3 cells is depicted in (**C**), and the migrated SCC-9 is depicted in (**D**). In (**E**), the quantification of the HSC-3 invasion is shown, and in (**F**), the invasion of SCC-9 cells. Values in graphics represent the mean ± SD of 3 independent experiments.

**Figure 7 cells-14-01311-f007:**
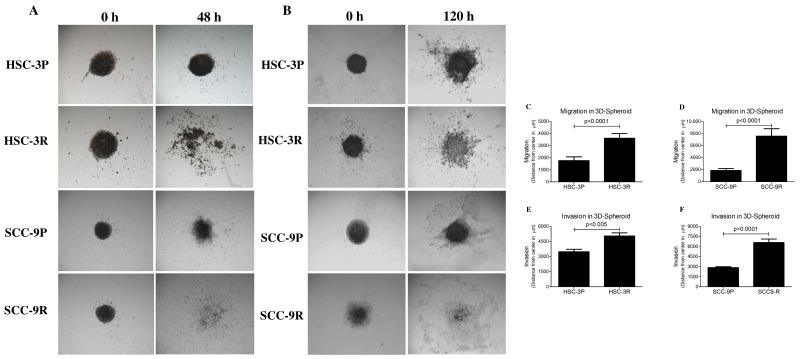
Cisplatin-resistant cells display increased migration and invasion in 3D spheroid models. (**A**) Representative images of the migration assay and (**B**) representative images of the invasion assay. Quantification of the 3D spheroid assays reinforces that resistant cells possess greater potential for migration and invasion than parental cells. The migration of HSC-3 cells and SCC-9 cells is depicted in (**C**) and (**D**), respectively, while their invasion is shown in (**E**) and (**F**), respectively. Values in graphics represent the mean ± SD of 3 independent experiments.

## Data Availability

The data supporting the findings of this study are included in the article and its Appendix A.

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
