# Peer review of "Generation and Characterization of Cisplatin-Resistant Oral Squamous Cell Carcinoma Cells Displaying an Epithelial–Mesenchymal Transition Signature"

_cells, 2025, doi:10.3390/cells14171311_

Round 1

Reviewer 1 Report

Comments and Suggestions for Authors

This study aimed to generate and characterize cisplatin-resistant oral squamous cell carcinoma (OSCC) models to elucidate resistance mechanisms. The Authors created two resistant OSCC cell lines (SCC-9R and HSC-3R) using a gradual dose escalation method. Analysis in both 2D and 3D cell culture models was conducted. Based on results, cisplatin resistance in OSCC involves epitelial-mesenchymal transition, inflammatory signaling, and metabolic adaptation.

These cisplatin-resistant cell lines can be used as valuable models for further investigations into strategies for reversing cisplatin resistance in OSCC, as well as for identifying the molecular mechanisms underlying chemoresistance.

The study is well-designed and described in the manuscript. Data generated is important for scientists working in the field of OSCC. However, before publication, several issues should be corrected:

  1. Fig. 1 - please add information about number of experimental repeats (3?). Which reagent was used to analyse viability in the 2D model?
  2. Fig. 2 - The manuscript contains a mirror image of Figure 2. Please correct.
  3. Lines 318-319 - "As shown in Fig. 4A, the migration of HSC-3R cells..." Please check and probably correct the number of Figure.
  4. Line 335 - "while their invasion is shown in (C) and (D), respectively." It should be (E) and (F).

Author Response

Manuscript ID: cells-3788397

Title: Generation and Characterization of Cisplatin-Resistant Oral Squamous Cell Carcinoma Cells Displaying an Epithelial-Mesenchymal Transition Signature

Dear Prof. Dr. Cord Brakebusch and Dr. Alexander E. Kalyuzhny,

Editor-in-Chief, Cells

We would like to thank the Reviewers for their valuable comments and suggestions for the manuscript cells-3788397. We responded to all the recommendations and, respectfully, we solicit the reevaluation of the revised manuscript for publication in the Cells.  

The comments and requests of the reviewers were carefully considered and followed by their respective answers below. All authors have seen and approved the changes (highlighted in red in the manuscript).    

Reviewer #1:

This study aimed to generate and characterize cisplatin-resistant oral squamous cell carcinoma (OSCC) models to elucidate resistance mechanisms. The Authors created two resistant OSCC cell lines (SCC-9R and HSC-3R) using a gradual dose escalation method. Analysis in both 2D and 3D cell culture models was conducted. Based on results, cisplatin resistance in OSCC involves epitelial-mesenchymal transition, inflammatory signaling, and metabolic adaptation.

These cisplatin-resistant cell lines can be used as valuable models for further investigations into strategies for reversing cisplatin resistance in OSCC, as well as for identifying the molecular mechanisms underlying chemoresistance.

The study is well-designed and described in the manuscript. Data generated is important for scientists working in the field of OSCC. However, before publication, several issues should be corrected:

Response:

We would like to thank you for providing us with timely and valuable comments and suggestions to improve the quality of manuscript.

  1. Fig. 1 - please add information about number of experimental repeats (3?). Which reagent was used to analyse viability in the 2D model?

Response:

We thank the reviewer for this observation. We have revised the legend of Figure 1 to indicate that all experiments were performed in triplicate. We have also specified that cell viability in both 2D and 3D models was assessed using the CellTiter-Glo 3D Cell Viability assay (Promega, Madison, Wisconsin, USA) according to the manufacturer’s protocol.

  1. Fig. 2 - The manuscript contains a mirror image of Figure 2. Please correct.

Response:

We thank the reviewer for pointing out this issue. The mirrored image in Figure 2 has been corrected in the revised version of the manuscript.

  1. Lines 318-319 - "As shown in Fig. 4A, the migration of HSC-3R cells..." Please check and probably correct the number of Figure.

Response:

We thank the reviewer for noticing this mistake. The figure citation has been corrected — the correct reference is to Figure 6A.

  1. Line 335 - "while their invasion is shown in (C) and (D), respectively." It should be (E) and (F).

Response:

We thank the reviewer for identifying this error. The text has been corrected to indicate the proper figure panels, which are (E) and (F).

Reviewer 2 Report

Comments and Suggestions for Authors

The manuscript from Dr Freitas de Morais et al. addresses a clinically relevant problem: cisplatin resistance in oral squamous cell carcinoma.

The generation and characterization of two cisplatin-resistant cell lines (SCC-9R and HSC-3R), combined with transcriptomic and functional analysis, represent a useful contribution and the use of 2D and 3D models and co-cultures with CAFs gives translational value to the work. Although there are previous studies on cisplatin resistance and EMT, the incorporation of multi-omics analyses and functional validations also support its originality.

This reviewer has minor suggestions for the authors:

1-Although EMT and resistance are correlated, no functional experiments (e.g., TWIST1 or ZEB1 knockdown) were performed to demonstrate direct connection. Please, discuss future directions to consolidate the functional role of these or another factors.

2- The supplementary figures are very useful, but some of them, for instance, Figure S3 (immunofluorescences) could be integrated into the main body of the manuscript (Figure 3) given their relevance.

3- In the manuscript (PDF), Figure 2 appears in an upside-down arrangement, I mean as if it were glued in a mirror image.

Author Response

Manuscript ID: cells-3788397

Title: Generation and Characterization of Cisplatin-Resistant Oral Squamous Cell Carcinoma Cells Displaying an Epithelial-Mesenchymal Transition Signature

Dear Prof. Dr. Cord Brakebusch and Dr. Alexander E. Kalyuzhny,

Editor-in-Chief, Cells

We would like to thank the Reviewers for their valuable comments and suggestions for the manuscript cells-3788397. We responded to all the recommendations and, respectfully, we solicit the reevaluation of the revised manuscript for publication in the Cells.  

The comments and requests of the reviewers were carefully considered and followed by their respective answers below. All authors have seen and approved the changes (highlighted in red in the manuscript).    

Reviewer #2:

The manuscript from Dr Freitas de Morais et al. addresses a clinically relevant problem: cisplatin resistance in oral squamous cell carcinoma.

The generation and characterization of two cisplatin-resistant cell lines (SCC-9R and HSC-3R), combined with transcriptomic and functional analysis, represent a useful contribution and the use of 2D and 3D models and co-cultures with CAFs gives translational value to the work. Although there are previous studies on cisplatin resistance and EMT, the incorporation of multi-omics analyses and functional validations also support its originality.

Response:

Thank you for highlighting the results of our study, and for the suggestions to improve it.

This reviewer has minor suggestions for the authors:

1-Although EMT and resistance are correlated, no functional experiments (e.g., TWIST1 or ZEB1 knockdown) were performed to demonstrate direct connection. Please, discuss future directions to consolidate the functional role of these or another factors.

Response:

We thank the reviewer for this valuable observation. We fully agree that functional studies are important to establish a direct causal link between EMT and cisplatin resistance. In response to your suggestion, we have expanded the final paragraph of the Discussion to highlight the need for functional validation through targeted modulation of EMT-related factors, as well as in vivo studies, to confirm the contribution of EMT to drug resistance and to guide the development of potential therapeutic strategies aimed at reversing EMT-induced chemoresistance in OSCC cell lines.

2- The supplementary figures are very useful, but some of them, for instance, Figure S3 (immunofluorescences) could be integrated into the main body of the manuscript (Figure 3) given their relevance.

Response:

We appreciate the reviewer’s suggestion and agree with the relevance of integrating Figure S3 into the main body of the manuscript. Accordingly, Figure S3 has been incorporated into Figure 4 in the revised version.

3- In the manuscript (PDF), Figure 2 appears in an upside-down arrangement, I mean as if it were glued in a mirror image.

Response:

We thank the reviewer for pointing out this issue. The arrangement of Figure 2 has been corrected in the revised version of the manuscript.